# Histological Effects of an Innovative 445 Nm Blue Laser During Oral Soft Tissue Biopsy

**DOI:** 10.3390/ijerph17082651

**Published:** 2020-04-13

**Authors:** Gaspare Palaia, Daniele Pergolini, Leonardo D’Alessandro, Raffaella Carletti, Alessandro Del Vecchio, Gianluca Tenore, Cira Rosaria Tiziana Di Gioia, Umberto Romeo

**Affiliations:** 1Department of Oral Sciences and Maxillofacial Surgery, Sapienza University of Rome, 00161 Rome, Italy; gaspare.palaia@uniroma1.it (G.P.); daniele.pergolini@uniroma1.it (D.P.); dalessandro.1634449@gmail.com (L.D.); alessandro.delvecchio@uniroma1.it (A.D.V.); umberto.romeo@uniroma1.it (U.R.); 2Department of Radiological, Oncological and Pathological Sciences, Sapienza University of Rome, 00161 Rome, Italy; carlettiraffaella@gmail.com (R.C.); cira.digioia@uniroma1.it (C.R.T.D.G.)

**Keywords:** blue laser 445 nm, oral biopsy, thermal effect

## Abstract

Continuously evolving laser devices can be used in various fields; they are an alternative to the traditional cold blade surgery to perform biopsies of oral soft tissues. The aspect focused on in this paper is the possibility to use the 445 nm diode laser (Eltech K-Laser srl, Treviso, Italy) in complete safety, by evaluating its thermal effects during microscopy. A histological evaluation of the alteration of the peri-incisional edges on 10 samples was realized. All excisional biopsies were related to clinically unsuspected lesions and performed by the same expert operator. The surgical procedure was performed with the same laser parameters and the same pathologist evaluated the thermal effect on the samples. An average value of the detected tissue alteration was calculated; the average damage of the epithelium was 650.93 μm, while in the connective tissue it was 468.07 μm. In all the cases a clear diagnosis was possible, and no clinical complications were observed; so, the 445 nm diode laser proved to be a device that can be safely used for biopsies of clinically unsuspicious lesions. Due to the small number of cases, this in vivo preliminary experience needs to be extended.

## 1. Introduction

Nowadays, the world trend in medicine is to follow scientific evolution, sustained by experimental research, in order to achieve and to prove the effectiveness of new devices that can simplify and make clinical activity more practical. In dentistry, laser is an innovative device, which has been used for several years but it is a current object of research and it is constantly evolving.

The first laser device dates back to 1960; it was invented by Theodore Maiman, and it was used in the dental field just four years later by R. Stern and R. Sognnaes. Lasers emit a typically monochromatic light with a specific wavelength that is highly focused, has a directional collimated beam, is evenly organized and efficient, and emitted like a coherent energy at high intensity [1,2].

Laser devices are used in all dental fields, from oral surgery to restorative dentistry, according to the features and type of wavelength. Recently, it has also found use in perioral aesthetics, such as in wrinkles or vascular lesions. Regarding oral soft tissue pathologies, the laser has found various uses over the years, as a device to make biopsy samples, or it can be used for the ablation of keratotic lesions or to photocoagulate vascular lesions or in photobiomodulation [1,2,3,4]. 

Regarding laser biopsies, a goal of this research is to analyze the thermal effects that occur at the peri-incisional tissues, following the surgical act of laser incision. In the surgical act, some lasers offer several advantages compared to the use of the scalpel [5]. On the oral soft tissues, based on the nature of the pathological tissue and the anatomical district, better results can be obtained by selecting the most suitable laser device. For example, a CO_2_ laser is optimal in the excision of very hydrated lesions, such as fibrous lesions, while diode lasers, which have a better coagulating action, can be exploited especially for very vascularized lesions or tissues, rich in vessels with a high amount of hemoglobin [6,7]. 

When a laser device is used in oral soft tissues surgery, many advantages could be described, such as easier surgical access, mostly in sites that are difficult to approach and do not offer a cleavage plan; better ergonomics [8]; accurate cut [9]; antimicrobial action of the surgical site [10]; more effective hemostasis, with intraoperative bleeding control and better visibility [5]; no sutures for intraoperative coagulation, with a second intention healing [6]; reduction of post-operative pain and edema [11]; reduction in the use of a local anesthetic [12]; reduced postoperative complications, such as dehiscence, and faster healing [7]; and faster and easier use [6]. By contrast, it requires a learning curve [13]. 

Among the possible disadvantages, in addition to the device’s cost, is the thermal effect of soft tissues being incised; this effect does not prevent having a histological diagnosis, but it can make reading of marginal tissues not optimal, in relation to the heat distributed to the cells impacted by the laser beam; so, alterations in the cell’s structure can be caused by the carbonization or denaturation of the proteins. Moreover, the evaluation about a complete excision of a pathological tissue at the level of the peri-incisional margins could be made hard, especially if the operator is not skilled [14,15]. 

Several studies have been made about the thermal effect on soft tissues of the oral cavity, both “ex vivo” and “in vivo”, with CO_2_ [6,16], diode [17], KTP [18], Nd:YAG [19] and Er:YAG [20] laser devices.

The blue diode laser is a novel device in the dental field; it is a semiconductor laser, with a GaN or InGaN active medium, emitting electromagnetic radiations in a range of 458–488 nm, perceived as blue by the human eye [21].

The aim of this study is to analyze the thermal effect determined by a blue light diode laser device (445 nm) of the last generation on oral soft tissues during a biopsy of unsuspected lesions, in order to define its use in excisional biopsies, as an alternative to the conventional scalpel method. 

## 2. Materials and Methods 

In this “case series”, patients were recruited in the Odontostomatologic Unit of Policlinic Umberto I of Sapienza University of Rome. 

Selection criteria were as follows: patients over 12 years of age; patients with systemic disease; patients with smoke addiction; unsuspicious oral soft tissue lesions; lesions not involving bone tissue; and clinically benign oral lesions within 2 cm in diameter. All patients with white and/or red patches or any ulcerated or erosive areas were excluded by the study.

Patients affected by oral vesiculobullous diseases, with a necrotic, exposed jaw bone or a history of local irradiation therapy, with oncological disease or with a severe state of immunosuppression were excluded in this study. 

This study was realized with a blue light diode laser (Eltech K-Laser srl, Treviso, Italy), with a wavelength of 445 nm ± 5 nm, at 2.5 Watt in Continuous Wave (CW), to make the surgical procedure faster, with a 320 μm surgical optical fiber and a fluence of 3100 J/cm^2^, according to the device settings. 

Local anesthesia, with 1.8 mL of mepivacaine solution (MEPIVACAINA PIERREL^®^, 30 mg/mL, injection solution 1.8 mL, Pierrel Spa, Milan, Italy) was administered before the beginning of each surgical intervention and after surgery all samples were sent to the pathologist for histological evaluation and diagnosis. Ten excisional biopsies of clinically benign oral lesions, of nine patients, as reported in Table 1, six women and three men, were performed in our Department, between November 2018 and January 2019. Two separate lesions were taken, during the same surgery, on the same woman patient. No sutures were applied to obtain wound healing by secondary intention.

All excised specimens were instantly fixed in a 10% neutral buffered formalin solution, then included in paraffin and stained with Hematoxylin and Eosin (H&E) for histological examination.

All patients were subsequently examined with 7 days and 1 month follow-ups. The prescribed medication for the patients was 0.2% chlorhexidine spray (Corsodyl^®^, GlaxoSmithKline Consumer Healthcare S.p.A, Baranzate, Milan, Italy) prescribed for three times a day for one week. No other medications (such as antibiotics or FANS) have been prescribed [22]. 

The histological analysis was performed with an optical microscope (Leica Leitz Camera; Leica Camera AG, Wetzlar, Germany). A computerized digital camera (Olympus Camedia 5050; Olympus Inc., Tokyo, Japan) was used to capture 5 Mp (24-bit color depth) images (×100 magnification) of the surgical resection margins (stored as JPG files). Computerized imaging software (ImageScope, Leica Biosystem, Milan, Italy) was utilized by a blinded pathologist to quantitatively evaluate the thermal effect in both the epithelium and connective tissue, expressed in microns.

The quantitative evaluation of the thermal effect at the peri-incisional margins on the histological section, stained with H&E, was measured with ImageScope software (ImageScope, Leica Biosystem, Milan, Italy). The depth of the tissue structural alterations, such as the coarctation and vacuolation of the squamous epithelium and the basophilic amorphous appearance of the connective tissue due to protein coagulation, were considered for the evaluation; the average thermal effects were calculated in epithelial and connective tissues, separately (Figure 1).

The statistical analysis, both for the mean and standard deviation, was done through the software Excel (Microsoft, Redmond, WA, USA), with several checks to identify any errors. In the Sample Standard Deviation formula, the numerator is the sum of the squared deviation of each specimen’s thermal effect, from the mean thermal effect, and the denominator is N − 1, where N is the number of specimens.

All the interventions were performed by the same oral surgeon and conducted according to and passing the online examination for the Certificate in Essential Good Clinical Practice (GCP) (Certificate number: EGCP19/3015) and following the guidelines of the World Medical Association, as well as adhering to the Declaration of Helsinki according to the local Ethical Committee guidelines (No. 4011). All participants underwent the same procedures without any deviation from the protocol during the entire trial and were informed about the methods of treatment and gave written informed consent.

## 3. Results

In all the cases, epithelial and connective thermal effects did not prevent specimen reading and the formulation of a histological diagnosis (Figure 2, Figure 3 and Figure 4); this according to the altered morphology and architecture of the tissue, due to the thermal effect, and which was evaluated by the pathologist.

The average thermal effect was 650.93 μm on epithelium, with a standard deviation of ±311.96, and 468.07 μm on connective tissue, with a standard deviation of ±264.23 (Figure 5). Only in one case there was an artifact of the epithelium, just over 1 mm, as reported in Table 2; the highest epithelial thermal effect was in Sample 5 while on the connective tissue it was in Sample 3. In all other cases, the thermal effect of both the epithelium and the connective tissue was lower than 1 mm, as reported in Table 2. The lesions that were identified as Pyogenic Granuloma during the histological examination, Samples 3 and 6.1 (Figure 2), showed the greatest thermal effect at the connective level compared to the other samples, due to the high vascular content of these lesions; this is because hemoglobin is the target chromophore for the 445 nm wavelength. The lesions that were identified, upon histological examination, as Squamous Papilloma, Samples 2 (Figure 3), 7 and 8 (Figure 4), showed a very low histological alteration relating to the heat, both at the level of the epithelium and of the connective tissue, due to a reduced vascular component of these HPV-related lesions.

Diode blue laser, in addition to guaranteeing an optimal intraoperative hemostatic effect, which allows the clinician to perform biopsies even in the most vascularized sites as soft palate, permitted an optimal reading of the specimen, as shown in these 10 biopsies.

## 4. Discussion

The importance of a biopsy is now proved; it is a surgical procedure that allows to have a diagnosis of certainty, which may or not confirm a clinical diagnostic suspicion [23]. Due to the important role of the biopsy, as the specimen taken is uniquely and thus not repeatable, even from a medical-legal point of view, this procedure needs to be performed within the full safety margins [14,15]. Since laser has been used in a versatile way as a useful device in the surgical field, even for biopsies, there have been several studies aimed to test its safety [10,18,20]; but there are several authors who, especially in the past, valued controversies in the use of these devices in the field of biopsy surgery [14,15,24]. The critical aspect is not the difficult diagnosis of the specimen, since the various laser devices are now able to guarantee a histological diagnosis, if used with the right parameters and by a skilled operator, but the legibility of the peri-incisional margins of the specimen. This is significant because the reading of the margins must be guaranteed in case of potentially malignant epithelial lesions, which can infiltrate tissues in depth and for which it is essential to understand if the pathological tissue has been completely excised [25,26]. In order not to run into this problem, the gold standard for suspicious lesions remains the cold blade, but alternatively the incision can be spaced, depending on the wavelengths used, at least 0.5 mm away from clinically visible pathological tissue [27,28]. Therefore, it is essential to correctly identify unsuspicious lesions by differentiating them from potentially malignant lesions, based on semiological criteria, identifying clinical diagnostic suspects that will be confirmed, or not, by histological examination; for this reason, clinicians should be qualified in oral pathology [29].

The possibility of having a diagnosis, with laser devices, is maintained, despite the thermal effect in the peri-incisional area due to the heat released by the photothermal effect, which increases the temperature of the tissue, at the incision point; so, it is clear that the laser can be used for biopsies of clinically benign lesions [30,31]. Compared to the traditional cold blade approach, however, the laser offers numerous advantages, including a hemostatic effect that defines intraoperative hemostasis, antibacterial effect on the incision path, better patient comfort, second intention healing without sutures and reduced postoperative edema [11]; for this reason, it is considered the gold standard in the approach to clinically benign lesions of oral soft tissues [32,33]. 

The histological study of the thermal effect is necessary to define the healthy tissue surrounding the lesion, which must be taken in order not to alter [6] the reading of the specimen, and be related to the specific laser device used, for its thermal effect, if it is used for biopsies of potentially malignant epithelial lesions [18,34].

Our results with a 445 nm blue laser, in addition to the intraoperative and postoperative advantages, agree with Reichelt et al. [35], who found no significant alterations at the level of the excised tissues, and all samples had a histological diagnosis; moreover, all the pieces showed the heat-related alterations were contained.

Comparing the results with a study with a blue laser of 445 nm “ex vivo”, on pig tongues, referring to the group of specimens excised with a power of 2 Watt in a continuous way (CW), the “in vivo” thermal effect was higher, both on the epithelium and connective tissue, but the epithelial thermal effect was mostly greater; furthermore, at various power and emission ways, the “ex vivo” thermal effect on the epithelium was greater than on the connective tissue, in all specimens [36].

The different results obtained “in vivo” are probably due to the vascular component, because the vascularization is reduced on a cadaver. Moreover more vascularized lesions, such as Pyogenic Granuloma, showed a greater thermal effect due to an increase in the laser parameters for intraoperative coagulation [37].

In fact, the lesions identified as Pyogenic Granuloma after histological examination, Samples 3 and 6.1 (Figure 2), showed the greatest thermal effect at the connective level compared to the other samples, due to the high vascular content of these lesions, because hemoglobin is the target chromophore for the 445 nm wavelength and because irradiation was repeated more frequently to obtain coagulation.

By contrast, the lesions that were identified, upon histological examination, as Squamous Papilloma, Sample 2 (Figure 3), 7 and 8 (Figure 4), showed very low histological alteration, relating to the heat, both at the level of the epithelium and of the connective tissue, due to a reduced vascular component of these HPV-related lesions and the poor intraoperative bleeding.

A limitation of the study was the reduced sample size, as such it is preliminary evidence; the objective is to increase the number of cases evaluated to have more indicative values related to oral soft tissue thermal effects.

Compared to the CO_2_ laser, the blue diode laser showed both lower epithelial and connective tissue average thermal effects [38].

The diode laser, compared to the cold blade or the Er: YAG laser [6,10,39], causes greater tissue thermal damage, but it still allows a good reading of the specimens. The ex vivo effect of the Er: YAG laser on soft tissues showed peripheral thermal damage using a power of 3.9 W, and remained within 1 mm, while with intermediate parameters the thermal effect was lower [20]; but the lack of tissues with blood flow in vivo, presented in our study, should be considered.

At present, few in vivo studies have been conducted with this wavelength, therefore it was not possible to compare the data obtained with similar devices [40,41,42].

## 5. Conclusions

As emerged from this preliminary study, in addition to the clinical advantages of laser devices, with a blue diode laser of 445 nm, an optimal evaluation of specimens and a histological diagnosis were allowed in all the cases.

Therefore, the laser confirms its use in the surgical field, as a tool for biopsies, if it is used for clinically benign lesions, for its contained thermal effect; however, a clinical evaluation of the lesions is required before the procedure.

For suspected lesions, the margins of tissue taken should be enlarged to healthy tissue, and in CW with this device may be taken with a safety margin of 1 mm, but other studies are needed to confirm that; in these cases, cold blade could be still considered the gold standard.

## Figures and Tables

**Figure 1 ijerph-17-02651-f001:**
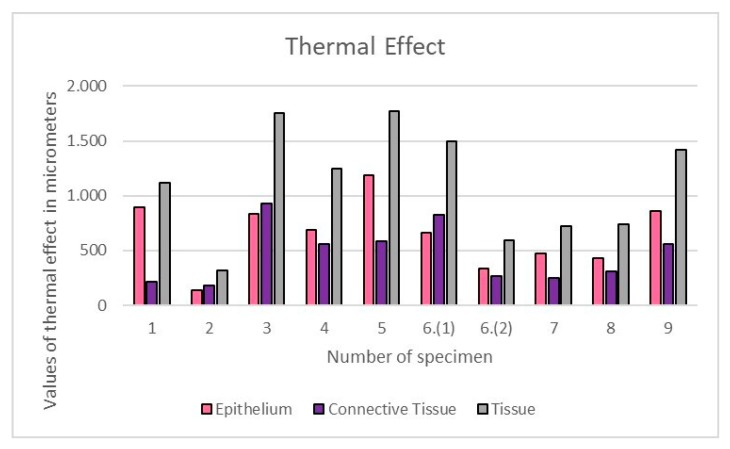
Values of thermal effects, expressed in μm, obtained on epithelium, connective tissues and in total for each specimen.

**Figure 2 ijerph-17-02651-f002:**
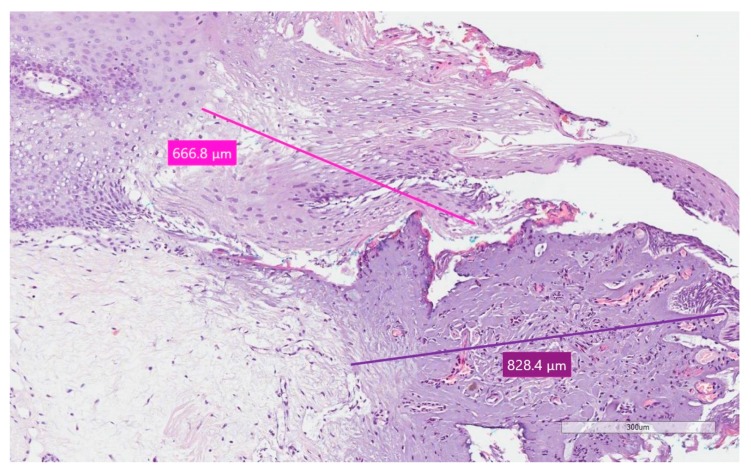
Specimen 1 of Case 6—Pyogenic Granuloma; stain H&E, magnification 10X. The pink line for epithelium and purple line for connective tissue point out the thermal effect.

**Figure 3 ijerph-17-02651-f003:**
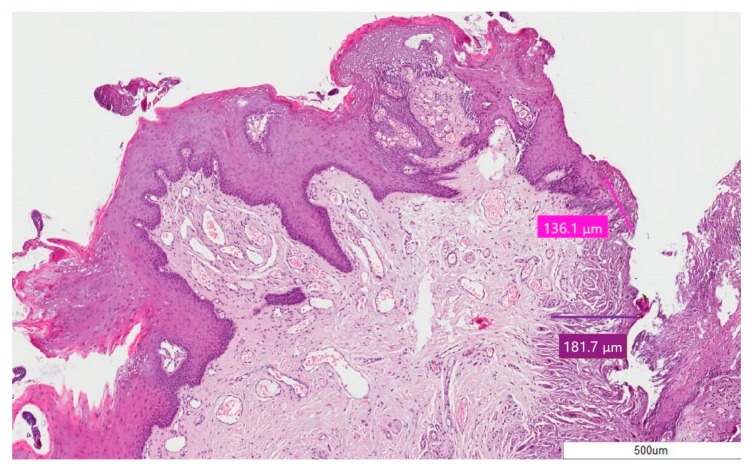
Case 2—Squamous Papilloma; stain H&E, magnification 5X. The pink line for epithelium and purple line for connective tissue point out the thermal effect.

**Figure 4 ijerph-17-02651-f004:**
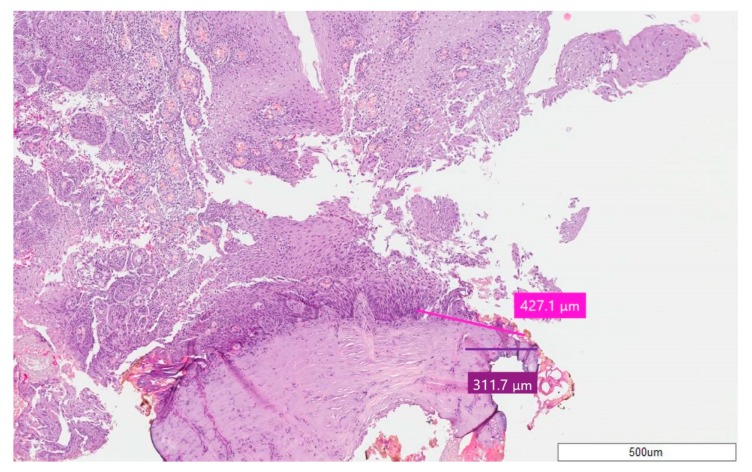
Case 8—Squamous Papilloma; stain H&E, magnification 5X. The pink line for epithelium and purple line for connective tissue point out the thermal effect.

**Figure 5 ijerph-17-02651-f005:**
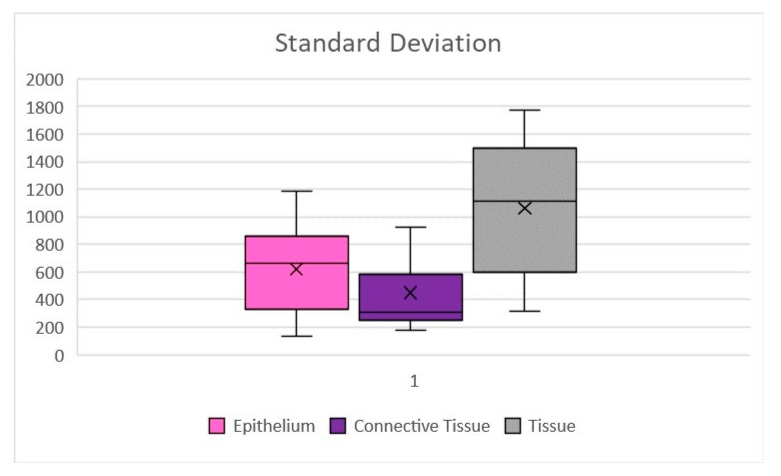
Sample standard deviations of the thermal effect on epithelial, connective tissue and complete tissue. Values expressed in μm.

**Table 1 ijerph-17-02651-t001:** Clinical history of the patients.

Patient	Sex	Age	Medical History	Drugs	Smoke	Site	Clinical Suspicion
1.	M	15	Nothing To Report	NO	NO	Vestibular interdental papilla	Pyogenic Granuloma
2.	F	63	Celiac’s Disease, Uterine Fibromas	NO	NO	Tip of the tongue	Iperplafocal Fibrous Hyperplasiasia
3.	F	42	Nothing To Report	Birth Control Pill	3 per day for 25 years	Keratinized Vestibular Gingiva	Pyogenic Granuloma
4.	M	43	Hypercholesterolemia, Kidney Stones, Psoriasis	Hypercholesterolemia Medication, Anti-Psoriatic	NO	Right buccal mucosa	Focal Fibrous Hyperplasia
5.	F	60	Hypertension	Antihypertensive Drug	25 per day for 47 years	Right buccal mucosa	Focal Fibrous Hyperplasia
6.	F	69	Nothing To Report	NO	NO	Tip of the Tongue (2 lesions)	Focal Fibrous Hyperplasia
7.	M	63	Pneumothorax, Inguinal Hernia	NO	10 per day for 30 years	Soft palate	Squamous Papilloma
8.	F	60	Hypertension, Sjogren’s syndrome	NO	NO	Hard palate	Squamous Papilloma
9.	F	72	Hypertension, Chronic Ischemic Heart Disease, Obesity	Antihypertensive Drugs, Diuretic	NO	Right buccal mucosa– right upper arch	Focal Fibrous Hyperplasia

**Table 2 ijerph-17-02651-t002:** The thermal effect obtained in each specimen for both epithelium and connective tissue.

Case	Histological Diagnosis	Epithelial Thermal Effect (μm)	Connective Tissue Thermal Effect (μm)	Tissue Thermal Effect (μm)
1.	Giant Cell Granuloma	898.7 ± 311.96	217.5 ± 264.23	1116.2 ± 505.37
2.	Squamous Papilloma	136.1 ± 311.96	181.7 ± 264.23	317.8 ± 505.37
3.	Pyogenic Granuloma	831.5 ± 311.96	926.9 ± 264.23	1758.2 ± 505.37
4.	Focal Fibrous Hyperplasia	690.8 ± 311.96	555.3 ± 264.23	1246.1 ± 505.37
5.	Focal Fibrous Hyperplasia	1188 ± 311.96	585.5 ± 264.23	1773.5 ± 505.37
6.(1)	Pyogenic Granuloma	666.8 ± 311.96	828.4 ± 264.23	1495.2 ± 505.37
6.(2)	Focal Fibrous Hyperplasia	331.7 ± 311.96	266.1 ± 264.23	597.8 ± 505.37
7.	Squamous Papilloma	477.1 ± 311.96	248.4 ± 264.23	725.5 ± 505.37
8.	Squamous Papilloma	427.1 ± 311.96	311.7 ± 264.23	738.8 ± 505.37
9.	Focal Fibrous Hyperplasia	861.5 ± 311.96	559.2 ± 264.23	1420.7 ± 505.37

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
