# Peer review of "Histological Effects of an Innovative 445 Nm Blue Laser During Oral Soft Tissue Biopsy"

_ijerph, 2020, doi:10.3390/ijerph17082651_

Round 1

Reviewer 1 Report

This study mainly evaluated thermal injury depth of oral biopsy specimens histologically using a new blue diode laser (445 nm wavelength).

Such studies are necessary with any new technology that induces thermal damage, but there are limitations that need to be addressed in the current manuscript as follows.

All of the evaluated lesions were benign or reactive, and not premalignant or malignant (by design apparently). Therefore, any conclusions regarding taking wider margins with laser (e.g. 1 mm as the authors propose in the conclusions section) are not supported by the current study and data and no references are given to support such a statement.

Further, a major issue with histopathologic diagnosis of lesions with thermal damage is not necessarily the depth of injury and tissue alteration, but the architectural and morphological changes to the affected tissues which can make histologic diagnosis challenging regardless of the depth of damage.

This is particular true for surgical or sectioned margins of dysplastic or potentially malignant lesions in addition to many other types of pathoses.    

Reviewer 2 Report

To the authors of article: "Histological effects of an innovative 445nm blue laser during oral soft tissue biopsy"

Please consider the following recommendations

On Page 2 – Line 61: Please add some references for diode lasers and their thermal effect on soft tissues.

On Page 2 – Line 64: I recommend that you rephrase the objective of the study in order to be more precise -" as an alternative to the conventional method with the scalpel", because the study presented in this article is not an comparative study.

On Page 3 – Line 117 : In Figure 1 on graph please insert the specimen no. 1 values and explain in the grid what represents the values in the Graph on horizontal please insert the subtitle (Number of specimens) and on vertical insert the subtitle (Values of thermal effects in micrometers)

On Page 3 – Line 120: In Figure 2 please mention and show on image what represent the values in micrometers (an area or a deep of thermal effect) and each values with different color (pink for epithelium and purple for connective tissues)

On Page 4 – Line 122: In Figure 3 please do the same as in Figure 2, explain and show what represents the values in image (with different colours for epithelium and connective tissues).

On Page 4 – Line 124: For Figure 4 the same, explanation and colors as in Figures 2 and 3.

On Page 4 –Line 126: In Figure 5 please also insert in the graph on the vertical grid the subtitle (Values in micrometers for the thermal effect)

On Page 5- Line 130: In Table 1 please explain what represents the acronym NTR? (Nothing To Report?)

On Page 6 – Line 141: In Table 2 the values are quite different, so I recommend for the next studies to increase the number of specimens in order to obtain a better statistical correlation.

On Page 6 – Line 170: Do you refer to the article of Reichelt or to the article of Fornaini et al. witch is [26] in the list of References?

References recommended for Introduction and Discussion:

  • Angiero, F., Parma, L., Crippa, R. et al.Diode laser (808 nm) applied to oral soft tissue lesions: a retrospective study to assess histopathological diagnosis and evaluate physical damage. Lasers Med Sci 27, 383–388 (2012). https://doi-org.am.e-nformation.ro/10.1007/s10103-011-0900-7
  • Jiang, D., Yang, Z., Liu, G. et al.A novel 450-nm blue laser system for surgical applications: efficacy of specific laser-tissue interactions in bladder soft tissue. Lasers Med Sci 34, 807–813 (2019). https://doi-org.am.e-nformation.ro/10.1007/s10103-018-2668-5
  • Braun, A., Kettner, M., Berthold, M. et al.Efficiency of soft tissue incision with a novel 445-nm semiconductor laser. Lasers Med Sci 33, 27–33 (2018). https://doi-org.am.e-nformation.ro/10.1007/s10103-017-2320-9
  • Vescovi P, Corcione L, Meleti M, Merigo E, Fornaini C,Manfredi M,cBonanini M, Govoni P, Rocca JP, Nammour S. Nd:YAG laser versus traditional A preliminary histological analysis of specimens from the human oral mucosa. Lasers Med Sci. 2010; 25(5):685–691.
  • Tuncer I, Ozçakir-Tomruk C, Sencift K, CöloÄŸlu S. Comparison of conventional surgery and CO2 laser on intraoral soft tissue pathologies and evaluation of the collateral thermal damage. Photomed Laser Surg 2010; 28(1):75-79.

Reviewer 3 Report

 Dear Authors,
This reviewer has read with interest your fine work. However, it is in my opinion, the manuscript has to be increased in some parts.

Materials and Methods
Line 70: You have omitted the number of approval by the ethical commette and it is necessary to publish researches on humans even if you reported to follow the local guidelines;

Line 93-97: please insert the type of statistic you used by means of the software. 

Results

Line 115, figure 1 please if possible enter the parameters you used to quantify the thermal effect;

figure 2, 3 and 4: it is necessary to insert a scale bar in each figure. A line is also needed to indicate the depth of the thermal effect in the epithelial and connective tissues in addition to the reported nms. In an attempt to better understand by the readers, I strongly suggest to magnify at least the microphotograph number 2 in which you reported the greater damages, in comparison to the other cases; 
Line 126, figure 5: it is necessary to specify the parameters you used to elaborate the standard deviation.

Discussion

Line 170: the reference you indicate as 26 doesn’t match with that of the reference list. Please revise the list.

Best regards 

Reviewer 4 Report

This article is about histological effects of an innovative 445nm blue laser
during oral soft tissue biopsy.

Here are a few comments that need to be addressed.
*P2, l47-53: there is a need for referencing each advantages cited in this paragraph.
* the introduction is well written, and interesting for a non-expert reader. However, the rationale to use a last generation blue light diode laser is unclear for the non-expert reader. Could the authors add some arguments/explanations regarding this innovative 445nm blue laser?
* It would be important to know in the introduction section if any other disadvantages than thermal effect on tissues is possible (both at the tissue and at the systemic level of the patient)
* P2, from line 67 to 72: it would be better to put this information on ethics at the end of the M&M section, since the reader can not be not familiar about "any deviation from protocol" at this stage.
* P2, l73-75: is there any reference, or rationale, to explain this device settings?
* P2. l78 "on A same woman patient".
* P2, l79-83: please describe the clinical procedure chronologically (i.e. describe the local anesthesia procedure before the description of the excisional biopsies.
* P2, l94: please indicate how many measures have been made for the quantitative evaluation of a biopsy (this number of evaluation will allow to present means and sd in Table 2)
* The type of study design should be reported in the M&M section (it looks like a case series analysis).
* Selection criteria for patients in this case series is important to give in order to be able to assess the external validity of this report. I understand it is a convenient sample of patients with clinically benign oral lesions, but it is important to give more precise information on selection criteria (in particular, non inclusion criteria are also important)
* P3, Figure 3.1: this type of graph is not relevant for this kind of data. Authors should use bars diagram, because there is no link between cases. Curves are interesting, for examples, for temporal trends.
* Table 1: what is the meaning of NTR
* Please make the sentence P6 l150-155 shorter, or split it in two sentences.
* The discussion section is well written, however, it would be interesting to have more indications (or references) on how to consider that a lesion is benign with certitude (which represents the main indication of lasers), given that the biopsy is also a way to determine if the lesion could be potentially malignant.
* P7 , l211 -213: limitations of the study shouldn't appear in the conclusion section.
